# Probiotic Supplements Effect on Feeding Tolerance, Growth and Neonatal Morbidity in Extremely Preterm Infants: A Systematic Review and Meta-Analysis

**DOI:** 10.3390/nu17071228

**Published:** 2025-04-01

**Authors:** Sofia Söderquist Kruth, Emma Persad, Alexander Rakow

**Affiliations:** 1Department of Women’s and Children’s Health, Karolinska Institutet, 17177 Stockholm, Sweden; sofia.soderquist.kruth@ki.se (S.S.K.); emma.persad@ki.se (E.P.); 2Women’s Health and Allied Health Professional Theme, Karolinska University Hospital, 17176 Stockholm, Sweden; 3Department of Neonatology, Karolinska University Hospital, 17176 Stockholm, Sweden

**Keywords:** probiotics, extreme prematurity, feeding tolerance, growth, necrotizing enterocolitis

## Abstract

Background/Objectives: Probiotic supplementation has been actively investigated in preterm populations to reduce the risk of necrotizing enterocolitis (NEC) and late-onset sepsis. Despite this, few studies have focused on clinically relevant feeding tolerance and growth outcomes, and there is an alarming lack of evidence surrounding extremely preterm infants (defined as birth before 28 weeks gestational age), those most at risk of severe comorbidities. We aimed to investigate whether probiotics improve feeding tolerance, neonatal growth and neonatal morbidity among extremely preterm infants. Methods: A literature search was conducted in Medline, Embase, Cochrane CENTRAL, Web of Science, and clinicaltrials.gov for ongoing trials. We included extremely preterm infants from randomized controlled trials and non-randomized trials with a concurrent control group. Two authors independently performed screening, data extraction and risk of bias assessment using the Risk of Bias 2 tool from Cochrane. The certainty of the evidence was assessed using GRADE. Results: Eleven RCTs and three non-randomized studies with a concurrent control group were included, analyzing a total of 14,888 extremely preterm infants. Meta-analyses revealed lower mean days to full enteral feeds (mean difference 1.1 days lower; 95% CI, 7.83 lower to 5.56 higher) and lower duration of parenteral nutrition (mean difference 2.4 days lower; 95% CI, 7.44 lower to 2.58 higher) in infants treated with probiotics; however, this was not statistically significant. There was a significant reduction in NEC (RR; 0.80, 95% CI; 0.68, 0.93) and all-cause mortality (RR; 0.56, 95% CI; 0.33, 0.93) in the probiotic group. All outcomes were graded at low or very low certainty of evidence. Conclusions: The findings indicate a trend towards a potential beneficial effect of probiotic supplementation in reducing feeding intolerance and a notable reduction of risk of NEC and all-cause mortality in infants receiving probiotics. Future RCTs will focus on feeding intolerance, and growth outcomes are warranted.

## 1. Introduction

Due to an immature gastrointestinal system and underdeveloped immune response, extremely preterm infants, defined as those born before 28 weeks of gestation, are at high risk of developing gastrointestinal-related complications, such as necrotizing enterocolitis (NEC), mortality [1,2] and late-onset sepsis (LOS) [3]. However, feeding intolerance and growth failure are two often-overlooked and more common complications, encompassing up to 75% of the neonatal population [4]. Feeding intolerance leads to delays or interruptions of enteral feeds, resulting in prolonged parenteral nutrition. This, in turn, increases the risk of line-associated infections and, subsequently, sepsis [5]. There is no unitary measure of feeding tolerance used in research or clinical practice; however, it is generally thought of as inability to tolerate desired enteral feeds. Clinical indicators of feeding intolerance can include gastric residuals, vomiting, poor intestinal mobility including delayed stool passage, and abdominal distension, and are frequently present with additional clinical manifestations, such as apnea and bradycardia, which often lead to the temporary suspension of enteral feeds to mitigate the potential development of NEC or LOS [6].

To date, the optimal timing to reestablish feeds following these fasting periods remains unclear, along with an understanding of the appropriate rate and volume for reintroduction [7,8]. A cautious approach is often taken, namely, a slow increase in enteral feeds; however, clinical practice can vary considerably between units. Unfortunately, this can prolong the length of time for which the infant receives inadequate nutrition, which may lead to growth failure, and this is still very common among extremely preterm infants [8,9]. This can cause long-term adverse health outcomes, as head circumference has been linked to neurological development, and weight and height gain is associated with improved lung function and lung maturation [10,11,12]. Optimizing the tolerance of enteral feeds and facilitating growth is thereby essential for the development of extremely preterm infants.

Due to the increased risk of gastrointestinal-related complications that preterm infants face during the neonatal period, probiotic supplementation has been investigated as a preventative measure. Probiotics are live microorganisms that can have numerous beneficial effects if administered in adequate numbers, including increasing the stimulation of immune functions, actively competing against pathogens, tightening the gut barrier, inducing the upregulation of cytoprotective genes, and degrading and fermenting certain foods and producing fat-soluble vitamins [13,14]. Many randomized controlled trials (RCTs) and systematic reviews have reported the effects of probiotic supplementation on NEC, LOS and death [15]. However, only a few have investigated the effects probiotics on feeding tolerance and growth [16,17,18,19], and those doing so typically involve infants born at a higher gestational age (GA) than extremely preterm infants [15]. As extremely preterm infants represent the neonatal patient group most vulnerable to feeding intolerance, growth failure and other adverse growth outcomes, there is an immense need for effective interventions to mitigate these complications. To our knowledge, this is the first systematic review reporting on the effects of probiotic supplementation on feeding tolerance and neonatal growth specifically in extremely preterm infants.

## 2. Materials and Methods

This systematic review has been constructed in accordance with the PRISMA guidelines for reporting systematic reviews and meta-analysis.

### 2.1. Eligibility Criteria

We limited our population to extremely preterm infants, defined as those born before 28 weeks GA. We also included studies investigating extremely low-birth-weight (ELBW) infants, provided average GA was specified in the study and the mean and standard deviation of GA for each group was below 28 weeks GA. We included RCTs and non-randomized trials with a concurrent control group. All eligibility criteria, containing inclusion and exclusion criteria for population, intervention, outcome and study design, are described in Appendix A.

### 2.2. Literature Search and Information Sources

A literature search up to 30 May 2024 was performed by an experienced information specialist in the following databases: Medline, Embase, Cochrane CENTRAL and Web of Science. In addition, the registry clinicaltrials.gov was searched for ongoing trials. To ensure that the searches were comprehensive, we conducted quality checks to identify known studies that should have been captured. All search terms are listed in Appendix A.

### 2.3. Search Strategy

The search strategy was developed in Medline (Ovid) in collaboration with librarians at the Karolinska Institute University Library. For each search concept, Medical Subject Headings (MeSH-terms) and free-text terms were identified. The search was then translated, in part using Polyglot Search Translator [20], into the other databases. No language restriction was applied and the search period was not limited. The strategies were peer-reviewed by another librarian prior to execution. De-duplication was performed using the method described by Bramer et al. [21]. A snowball search was applied to check the references and citations of eligible studies from the database searches using Covidence online software https://www.covidence.org/ (29 May 2024).

### 2.4. Data Collection and Selection Process

Two team members (S.S.K., E.P.) independently reviewed all titles and abstracts for eligibility against our inclusion/exclusion criteria using the Covidence online software. All studies included in the title and abstract screening phase were screened at the full text level independently, using the same inclusion/exclusion criteria. If any disagreements arose during the screening process, they were resolved by discussion and consensus or by consulting a third member of the team. We recorded the exclusion reason for each full text and a comprehensive list of excluded studies is displayed in Appendix A.

### 2.5. Data Extraction

Data from studies meeting our inclusion criteria were extracted into evidence tables by two reviewers independently (S.S.K., E.P.). We extracted the characteristics of study populations, settings, interventions, comparators, study designs, methods, results, and information about funding source.

### 2.6. Data Items

#### 2.6.1. Primary Outcome Measures

Feeding tolerance, with the following outcome measures:
○Average enteral volume (first 28 days);○Duration of parenteral nutrition;○Interruptions of enteral feeds (number of days/episodes);○Number of gastric residuals per week;○Time to full enteral feeds (120–160 mL/kg/day);○Time to 100 mL/kg/day enteral feeds;○Time to 50 mL/kg/day enteral feeds.Growth, with the following outcome measures:
○Growth velocity—weight gain, length gain, increase in head circumference during neonatal period;○Weight at end of neonatal period (gestational week 36–40);○Length at end of neonatal period (gestational week 36–40);○Head circumference at end of neonatal period (gestational week 36–40);○Standard deviation score during neonatal period and at end of neonatal period (gestational week 36–40);○Weight > 10 percentile at gestational week 34;○Growth failure (other definition).

#### 2.6.2. Secondary Outcome Measures

Necrotizing enterocolitis Bell Stage II–III;Sepsis (late-onset sepsis and culture-proven late-onset sepsis);All-cause mortality (during hospitalization);Length of hospitalization;Adverse events—any reported adverse events related to the intervention of probiotic supplements.

### 2.7. Missing Data

Some trials included both very and extremely preterm infants, but did not separate these groups when reporting outcomes. From January to July 2024, we requested data for extremely preterm infants from authors. A clarification of missing or unclear outcome reporting for important variables was also requested. Out of 29, 6 responded, and 4 got back to us with data [22,23,24,25]. We also contacted authors with trial registration and conference abstracts about data, but received no answers.

### 2.8. Effect Measures

We calculated risk ratios (RR) for dichotomous data and mean differences (MD) for continuous data, including the 95% confidence interval for all variables included in meta-analysis.

### 2.9. Synthesis Methods

For all outcome variables where at least three studies had reported on the outcome, we conducted meta-analyses using Review Manager (RevMan) software version 8.8.0. For outcomes, the variables were fewer than three studies having reported on the outcome, and data were narratively synthesized by describing the results of each study and comparing them qualitatively.

#### 2.9.1. Heterogeneity

To measure outcomes on which we performed meta-analyses, we evaluated heterogeneity by assessing the sizes and positions of the confidence intervals in forest plots. Heterogeneity was also assessed statistically using Cochran’s Q Test. An I^2^ over 50% was considered moderate heterogeneity and over 70% was considered substantial heterogeneity.

#### 2.9.2. Subgroup Analysis

Subgroup analysis was performed assessing RCTs and non-randomized studies separately.

#### 2.9.3. Sensitivity Analysis

To test the robustness of our results, we conducted a number of sensitivity analysis. These included meta-analyses with only low risk of bias studies and meta-analyses only including studies with the largest sample sizes to see if the conclusions remained consistent with only the most heavily weighted studies. We also performed all meta-analyses with both random-effect and fixed-effect models. Furthermore, sensitivity analyses removing studies without a clear definition and/or slightly different definitions than other studies (growth at discharge and sepsis) were conducted.

### 2.10. Risk of Bias Assessment

Cochrane Risk of Bias Tool 2.0 was used to assess risk the for randomized trials [26]. For non-randomized studies, we used ROBINS-I (Risk of Bias in Non-randomized Studies of Interventions) [27]. Two independent reviewers assessed the risk of bias for each study. Disagreements in the risk of bias assessment were resolved through discussion and consensus or by consulting a third reviewer.

### 2.11. Certainty Assessment

We graded the strength of evidence based on the guidance established by the GRADE (Grading of Recommendations Assessment, Development and Evaluation) Working Group [28]. Two reviewers assessed each domain for each key outcome together, and any discrepancies were resolved by consensus. We graded the certainty of evidence for the outcomes deemed to be of greatest importance to decision-makers and those commonly reported in the literature by carefully considering the ratings of each domain.

## 3. Results

### 3.1. Results of the Search

The searches generated 2040 titles and abstracts. Following de-duplication, 2023 underwent title and abstract screening, of which 1650 studies were excluded. After a full text review of 373 studies, a final set of 28 studies comprising 14,888 extremely preterm infants was included. Reasons for exclusion at the full text level are summarized in the PRISMA flowchart (Figure 1).

### 3.2. Study Design

A description of patient demographics, study setting, and the design of the included and ongoing studies is found in Table 1, Table 2 and Table 3. We included eleven RCTs and three non-randomized studies with a concurrent control, one of which was labeled as awaiting classification [22,23,24,25,29,30,31,32,33,34,35,36,37,38]. We also included 14 ongoing trials. Two of the studies reported on patients derived from the same sample population but reported on different feeding outcomes [29,34]. To avoid duplication, we only evaluated data on NEC, LOS, death, and hospitalization from the original article [29]. The studies were conducted across eleven countries. Ten of the studies were multicenter and four were single-center.

**Figure 1 nutrients-17-01228-f001:**
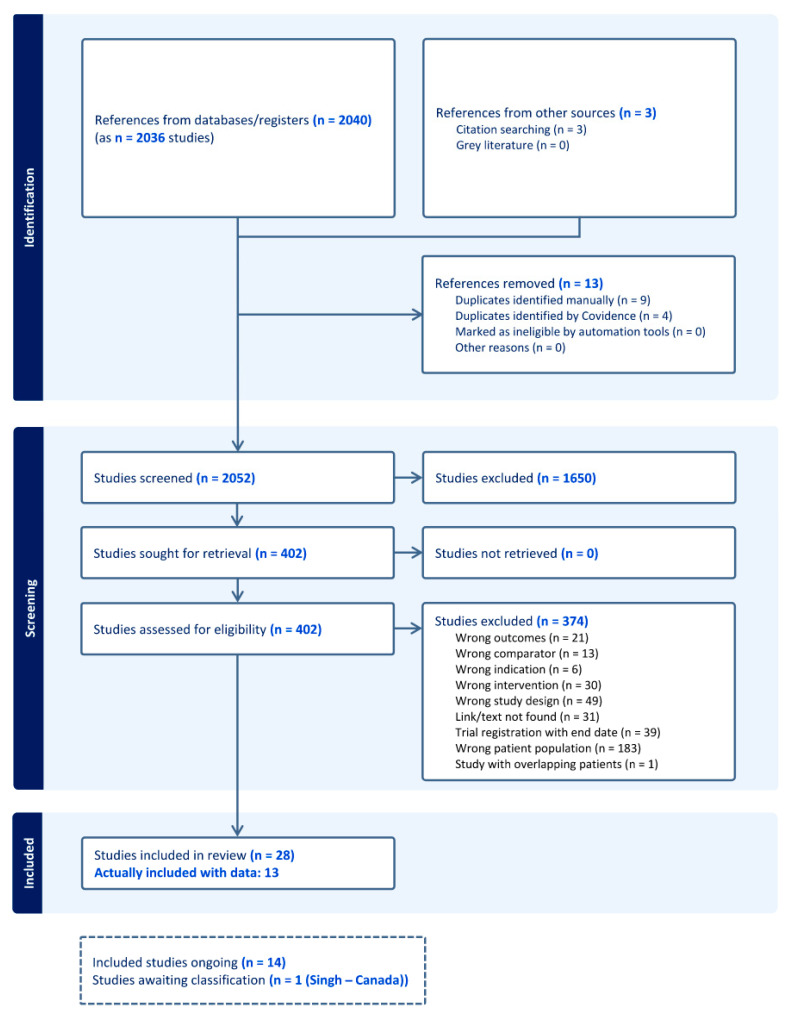
PRISMA flowchart of screened, excluded and included studies.

### 3.3. Study Population

All studies included data for infants < 28 weeks GA. Four of the full-text studies only reported on infants under <28 weeks GA or birth weight < 1000 g with a mean GA and SD < 28 weeks GA [29,30,31,34]. For the other nine full-text studies, infants < 28 weeks GA or birth weight < 1000 g were reported as a subgroup [22,23,24,25,32,33,35,36,37]. For four of the studies, we requested and obtained data for infants born <28 weeks’ GA not available in the original article [22,23,24,25]. The GA spanned from week 22 to 27. Only one study included infants born at 22 weeks GA [25]. Three studies included from 23 weeks GA [24,32,36,38], and one from 24 weeks GA [22]. The other nine studies did not report on the range of GA at birth [23,29,30,31,33,34,35,37]. The sample size of extremely preterm infants in each study is presented in Table 1.

### 3.4. Study Interventions

Six studies compared probiotics to placebo [24,25,32,35,36,38], and eight compared probiotics to no treatment [22,23,29,30,31,33,34,37]. Two studies did not involve a placebo but were still blinded as the study intervention was mixed in separate units [29,34]. Two of the trials were open-label [30,33] and one had no information about blinding [22]. Additionally, three of the studies were non-randomized cohort studies [23,31,37].

There was a large variability in type of probiotics supplemented. Only four studies had the same probiotic or probiotic combination as another study. Six studies used single-strain probiotics [24,25,32,33,36,38] and eight used multi-strain probiotics [22,23,29,30,31,33,34,36]. For the studies with single-strain probiotics, two studies used *Lactobacillus reuteri* [33,38], one used *Bifidobacterium breve M-16V* [24], one used *Bifidobacterium breve BBG-001* [32], another used *Bifidobacterium bifidum OLB6378* [25], and one used *Lactobacillus rhamnosus GG* with a combination of bovine lactoferrin [36]. Of the studies with multi-strain probiotics, two studies had the same combinations of *Lactobacillus rhamnosus GG* and *Bifidobacterium infantis* [29,34], whereas for the other studies, the combinations of probiotics differed and the number of probiotic strains varied between two and five strains. Some probiotic strains were used in more than one study. Six of the studies included *Lactobacillus rhamnosus GG* [22,29,30,34,36,37], five studies included *Bifidobacterium infantis* [23,29,34,35,37], three studies included *Bifidobacterium bifidum* [25,30,31], and two studies included *Bifidobacterium longum* [30,37]. A detailed description of the type of probiotics and placebo used in every study is found in Table 1.

### 3.5. Risk of Bias in Included Studies

An overview of the methodological quality of the included studies is presented in Appendix A. RoB was assessed with the Cochrane Risk of Bias Tool 2.0 for RCTs and the ROBINS-I tool for non-randomized studies. Out of the eleven included RCTs, five where rated at an overall low RoB. Five were rated at an overall moderate RoB and one was rated as high risk of bias. Of the two non-randomized studies, one was rated at a moderate RoB and one at serious overall RoB due to risk of confounding and potential bias in outcome measurements.

### 3.6. Certainty of the Evidence

Two review authors independently assessed the certainty of evidence for all outcomes, where we performed meta-analyses including days to full enteral feeds, days with parenteral nutrition, weight at discharge, length at discharge, head circumference at discharge, NEC, late-onset sepsis, all-cause mortality and hospitalization. All outcomes except days to full enteral feeds included data from non-randomized studies, automatically downgrading the outcomes by one. For our primary outcomes of feeding tolerance and growth, all outcomes except head circumference at discharge were downgraded for inconsistency, indirectness and imprecision. For our secondary outcomes, all outcomes were downgraded for at least one reason out of inconsistency, indirectness and/or imprecision. For the outcomes where no visual or statistical issues with heterogeneity were detected, we did not downgrade for inconsistency. All outcomes were graded at either low or very low certainty of evidence.

### 3.7. Main Results

In Table 4, all the outcomes and main clinical findings are listed from every study, respectively. Table 5 provides a summary of the main results, including certainty of evidence. Figure 2, Figure 3, Figure 4, Figure 5, Figure 6, Figure 7, Figure 8, Figure 9 and Figure 10 show a meta-analysis of days to full enteral feeds (150 mL/kg/day), days with parenteral nutrition, growth (weight, length and head circumference) at discharge, NEC, late-onset sepsis, all-cause mortality and hospitalization. All other reported outcomes are narratively summarized.

**Figure 2 nutrients-17-01228-f002:**
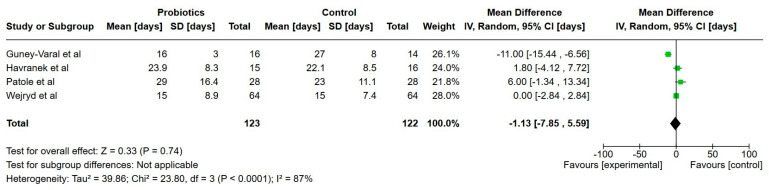
Forest plot of the effect of probiotics supplements vs. no probiotics on time to full enteral feeds (150 mL/kg/day). Results are presented per study and pooled analysis in bold.

**Figure 3 nutrients-17-01228-f003:**
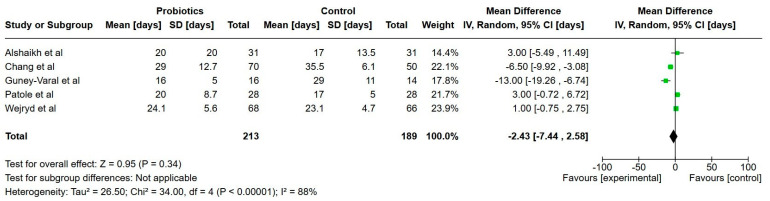
Forest plot of the effect of probiotics supplements vs. no probiotics on duration of parenteral nutrition (number of days). Results are presented per study and pooled analysis in bold.

**Figure 4 nutrients-17-01228-f004:**
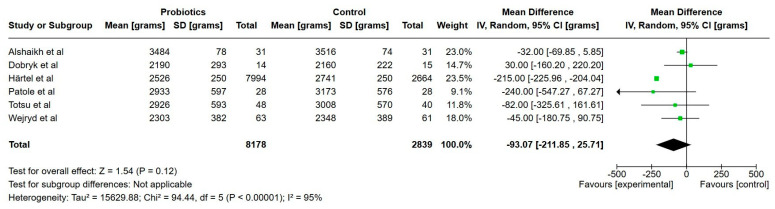
Forest plot of the effect of probiotics supplements vs. no probiotics on weight at discharge from neonatal unit. Results are presented per study and pooled analysis in bold.

**Figure 5 nutrients-17-01228-f005:**
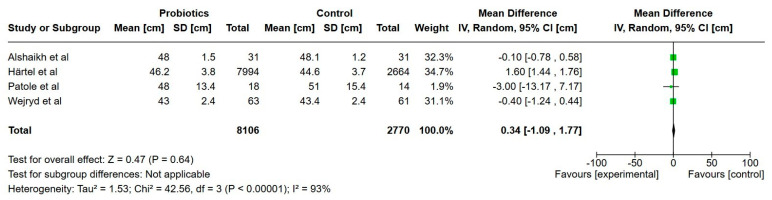
Forest plot of the effect of probiotics supplements vs. no probiotics on length at discharge from neonatal unit.

**Figure 6 nutrients-17-01228-f006:**
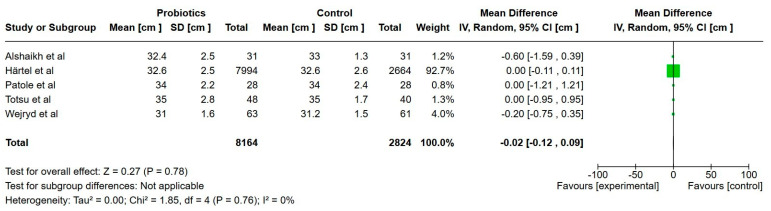
Forest plot of the effect of probiotics supplements vs. no probiotics on head circumference at discharge from neonatal unit. Results are presented per study and pooled analysis in bold.

**Figure 7 nutrients-17-01228-f007:**
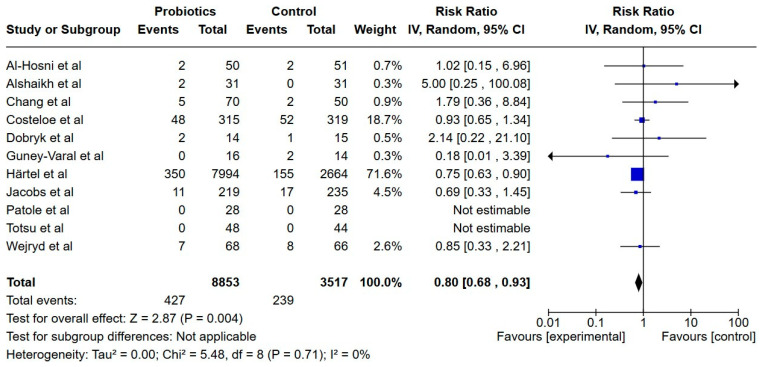
Forest plot of the effect of probiotics supplements vs. no probiotics on necrotizing enterocolitis. Results are presented per study and pooled analysis in bold.

**Figure 8 nutrients-17-01228-f008:**
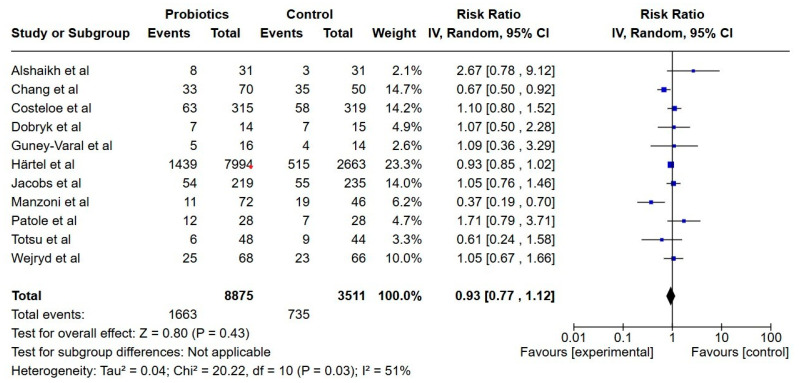
Forest plot of the effect of probiotics supplements vs. no probiotics on late-onset sepsis. Results are presented per study and pooled analysis in bold.

**Figure 9 nutrients-17-01228-f009:**
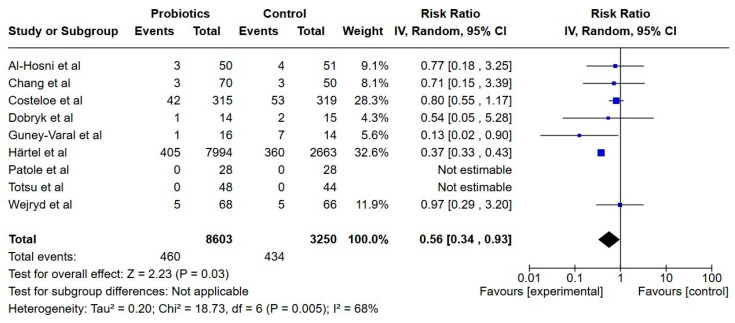
Forest plot of the effect of probiotics supplements vs. no probiotics on all-cause mortality. Results are presented per study and pooled analysis in bold.

**Figure 10 nutrients-17-01228-f010:**
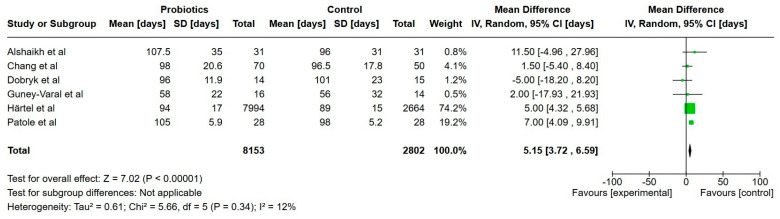
Forest plot of the effect of probiotics supplements vs. no probiotics on hospitalization. Results are presented per study and pooled analysis in bold.

#### 3.7.1. Primary Outcomes

##### Feeding Tolerance

Nine studies reported on our primary outcome of feeding tolerance [22,24,25,29,30,31,33,34,38]. In total, there were twelve different feeding tolerance outcomes reported, including the following: average feeding volume, days with interrupted feeding, duration of parenteral nutrition, enteral nutrition interrupted for over 24 h, episodes of feeding intolerance, feeding intolerance episodes numbering three or more, gastric residuals (larger than 2 mL/kg and exceeding the volume of the previous meal), time to enteral nutrition of 50 mL/kg/day, time to enteral nutrition of 100 mL/kg/day, time to full enteral nutrition (120 mL/kg/day), time to full enteral nutrition (150 mL/kg/day), and time to full enteral nutrition (160 mL/kg/day).

Four studies reported on time to full enteral feeds (150–160 mL/kg/day) and included a total of 251 extremely preterm infants [22,24,33,34]. The mean times to full enteral feeds were 1.1 days shorter in the probiotics group but not statistically significant (95% CI, 7.83 lower to 5.56 higher), with a very low certainty of evidence.

Five studies reported on the duration of parenteral nutrition and included a total of 402 extremely preterm infants [22,24,30,31,38]. The duration of parenteral nutrition was 2.4 days shorter in the probiotics group but not statistically significant (95% CI, 7.44 lower to 2.58 higher), with a very low certainty of evidence.

Of the seven studies reporting on the narratively summarized feeding tolerance outcomes [22,24,25,29,30,33,38], there were two studies that found some significant effects on feeding tolerance in favor of probiotics [24,30]. Four studies found no significant effect [25,29,33,38], and one study reported that probiotics had no or exacerbated the effects for one feeding outcome [24]. One study found a significant reduction in days to full enteral feeds defined as 120 mL/kg/day for the probiotic group, but no effect on interruption of enteral feeds due to feeding intolerance [30]. Another study found that probiotic supplements significantly decreased days to enteral feeds of 100 mL/kg/day and reduced feeding intolerance episodes, defined as abdominal distension, gastric residue or vomiting [22]. For the studies with no significant beneficial effect with probiotics on feeding tolerance, one study reported on two feeding outcomes: days to full enteral feeds, defined as reaching 160 mL/kg/day and number of episodes with reduced feeding tolerance, finding fewer days to full enteral feeds and fewer episodes of reduced feeding tolerance, although these were not statistically significant [33]. Another study reported on the average volume of feeding during the first 28 days and found no significant effect of probiotic treatment [29]. However, in subgroup analysis, they observed that the average daily feed was significantly higher in the control group for infants with birth weights of 751–1000 g. This result was not seen in infants with birth weights of 500–750 g. One study reported on days to 50 and 100 mL/kg/day and found no significant effect between the probiotic and control groups for these levels of enteral intake [24]. However, there was a significant difference favoring the control group at 150 mL/kg/day. Another found no difference between groups for the outcome time to 100 mL/kg/day [25], and one found no significant difference between the probiotics and control group for the outcomes days with interrupted feeding and numbers of gastric residuals [38].

##### Growth

Seven studies reported on our primary outcome of growth [23,24,25,29,30,33,38]. In total, there were 26 different growth outcomes reported, including the following: change in weight z-score between birth and 40 weeks GA, growth velocity (grams/kg/day), growth velocity day 7–36 weeks GA, growth retardation at discharge, head circumference day 14 and 28, head circumference at 40 weeks GA, head circumference Z-score at 40 weeks GA, head circumference at discharge, length day 14 and 28, length at 36 weeks GA, length at 40 weeks GA, length Z-score at 40 weeks GA, length at discharge, weight day 7, 14, 21 and 28, weight at 34 weeks GA, weight at 36 weeks GA, weight at 40 weeks GA, weight at discharge, weight < 10 percentile at 34 weeks GA, weight < 10 percentile at 40 weeks GA and weight Z-score at 40 weeks GA.

For the majority of outcomes, only one or two studies reported on the same outcome. For many variables, studies used similar but not the exact same definitions of outcomes. We conducted meta-analyses for growth at discharge, including weight, length and head circumference. Some of the studies reported growth at discharge, whereas some reported growth at 36 or 40 weeks GA. We assumed that discharge took place around 36–40 weeks GA, and therefore included all these outcome variables in the meta-analysis for growth at discharge.

Six studies reported on weight at discharge and included a total of 10,988 extremely preterm infants [23,24,25,30,33,38]. The mean weight at discharge was 88 g lower in the probiotics group, but this was not statistically significant (95% CI, 205 lower to 30 higher), and the certainty of evidence was very low.

Four studies reported on length at discharge and included a total of 11,068 extremely preterm infants [23,24,30,38]. The mean length at discharge was 0.3 cm higher than the probiotic group, but this was not statistically significant (95% CI, 1.1 lower to 1.8 higher), and the certainty of evidence was very low.

Five studies reported on head circumference at discharge, including a total of 10,876 extremely preterm infants [23,24,25,30,38]. The mean head circumference at discharge was 0.02 cm lower in the probiotics group, but this was not statistically significant (95% CI, 0.12 lower to 0.09 higher), and the certainty of evidence was low.

Four studies also reported on other growth outcomes additional to those included in the meta-analysis [24,29,30,38]. In the narratively summarized growth outcomes, there were three studies that found some significant beneficial effects on growth with probiotics [29,30,38], and one study that found an increased rate of impaired growth in infants treated with probiotics [24]. One study found that infants treated with probiotics had a significantly higher growth velocity [29]. However, in the subgroup analysis, it was observed that growth velocity and average daily weight gain were significantly improved in infants with birth weight 500–750 g, but no effect was seen on infants with birth weight 751–1000 g. The authors found no effect on weight for infants < 10 percentile at 34 weeks GA between the probiotic and control group. Another study reported a significantly improved growth velocity in the probiotics group between day-seven and 36-week GA, but found no other significant effects for the other growth outcomes [30]. In another study, the authors reported weight, length and head circumference for days 14 and 28 [38]. In addition, weight was also reported for days 7 and 21. The study did not find any significant results that probiotics effect weight or length. However, improved head growth was observed during the first month of life. The final study reported on growth retardation at discharge, finding 67.8% of infants in the probiotics group had impaired growth compared to 35.7% in the control group, which was statically significant [24]. However, no clear definition of growth retardation was provided.

#### 3.7.2. Secondary Outcomes

##### Necrotizing Enterocolitis

Eleven studies reported on NEC, including a total of 12,369 extremely preterm infants [22,23,24,25,29,30,31,32,33,35,37]. The NEC incidence was lower in the probiotics group (RR; 0.80, 95% CI; 0.68, 0.93) and the certainty of evidence was low.

##### Late-Onset Sepsis

Eleven studies reported on late-onset sepsis, including a total of 12,452 extremely preterm infants [22,23,24,25,30,31,32,33,35,36,38]. We only included studies that used the definition late-onset sepsis or culture-proven sepsis. The incidence of late-onset sepsis was lower in the probiotics group, but it was not statistically significant (RR; 0.94, 95% CI; 0.79, 1.11), and the certainty of evidence was low.

##### All-Cause Mortality

Nine studies reported on all-cause mortality during hospital stay, including a total of 11,853 extremely preterm infants [22,23,24,25,29,31,32,33,38]. The incidence of all-cause mortality was lower in the probiotics group (RR; 0.56, 95% CI; 0.33, 0.93) and the certainty of evidence was very low.

##### Hospitalization

Five studies reported on hospitalization, including a total of 10,887 extremely preterm infants [22,23,24,31,33]. The mean days of hospitalization was 5 days higher in the probiotics group, and was statistically significant (95% CI, 3.7 to 6.6 higher). The certainty of evidence was graded as very low.

##### Adverse Events

None of the studies reported any adverse events related to the probiotic supplementation.

##### Results of Subgroup and Sensitivity Analysis

To test the robustness of our results, we performed subgroup analyses with different study designs by testing RCTs and non-randomized studies separately. The biggest effects when removing non-randomized studies were seen for duration of parenteral nutrition, NEC and all-cause mortality. For duration of parenteral nutrition, the exclusion of non-randomized studies reduced the mean difference from 2.4 to 1.25 days lower for the probiotics group, but the results were still not statistically significant. For NEC, the risk ratio decreased, and the risk reduction seen in the original analysis was no longer statistically significant (RR; 0.89, 95% CI; 0.66, 1.20). The same was observed for all-cause mortality (RR; 0.76, 95% CI; 0.54, 1.07). For hospitalization, the mean difference in days was similar but no longer statistically significant (mean difference; 4.67, 95% CI; 1.76 lower to 11.09 higher). Results from other meta-analyses show negligible differences when removing non-randomized studies.

We also tested if our result differed when only evaluating studies with low risk of bias. For our primary outcomes, days to full enteral nutrition and days with parenteral nutrition, we observed a change in direction of risk ratio, with the mean difference now being two days higher in the probiotics group (mean difference; 2.0, 95% CI; 3.5 lower to 7.5 higher) for days to full enteral feeds, and 1.4 days higher for days with parenteral nutrition for the probiotics group (mean difference; 1.4, 95% CI; −0.22 lower to 2.9 higher), but this was still not statistically significant for any of the outcomes. Also, only two studies could be included as studies with low risk of bias for the assessment of these outcomes. For growth at discharge, the results remained consistent with our primary analysis. For NEC, the risk ratio remained similar when only including low-risk-of-bias studies, but the results were no longer statistically significant. For all-cause mortality, the risk difference between probiotics and control was smaller and no longer statistically significant. For sepsis and hospitalization, the results remained consistent with those in our primary analysis. Further, we had some concerns with missing data for some of the studies [25,34]. However, when removing these studies in the meta-analysis, this had little to no effect on the synthesized results.

Most studies investigated a similar number of extremely preterm infants, ranging from 15 to 70 infants. However, one of the cohort studies was a heavily weighted, country-wide analysis with a sample size of 10,658 extremely preterm infants. When removing this study, the risk reductions of NEC and all-cause mortality were no longer significant for the probiotics group. Likewise, the outcome of increased length of hospitalization was no longer statistically significant for the probiotics group. Sensitivity analyses for other outcomes were comparable to the primary analysis. We also performed sensitivity analysis to test if our results remained consistent when using a fixed-effect model or random-effect model. For the outcome days to full enteral feeds, the mean difference changed from 1.1 days lower in the probiotics group (95% CI, 7.83 lower to 5.56 higher) to 1.7 days lower (95% CI, -3.82 lower to 0.36 higher). Days with parenteral nutrition changed from 2.4 days lower in the probiotics group (95% CI, 7.44 lower to 2.58 higher) to 0.6 days lower (95% CI, 1.96 lower to 0.81 higher). The mean weight at discharge was on average 200 g lower in the probiotics group (95% CI, 208 lower to 187 g lower), and the mean length at discharge was 1.4 cm higher in the control group (95% CI, 1.28 higher to 1.60 higher). However, for these outcomes, the use of random effect models was deemed appropriate due to the large heterogeneity. For NEC, sepsis, all-cause mortality, hospitalization, and head circumference at discharge, the results were similar with a random or fixed effects model.

Further, we wanted to investigate if our results differed when only including studies with the definition late-onset sepsis, thereby removing studies reporting on culture-proven sepsis. However, this change did not alter the results. Furthermore, we removed studies with growth data at 36- and 40-week GA as measures for growth at discharge. For head circumference, this did not alter the results. However, for weight, the mean difference was 215 g lower in the probiotics group (95% CI, 226 lower to 204 lower), and for length, the mean difference was 1.6 cm higher in the control group (95% CI, 1.4 higher to 1.8 higher).

## 4. Discussion

### 4.1. Summary of Main Results

This systematic review and meta-analysis investigated the effects of probiotics supplementation during the neonatal period on feeding tolerance, growth and neonatal morbidity among 14,888 extremely preterm infants. Overall, we found limited evidence that probiotics improved feeding tolerance. Despite this, we noted a favorable, non-significant trend for the outcomes mean days to full enteral nutrition and days with parenteral nutrition, whereby the probiotics group exhibited lower days for these outcomes. Furthermore, some of the studies with narratively summarized feeding outcomes found some beneficial effect of probiotic supplementation on feeding tolerance. For the growth outcomes in our meta-analyses, we found no evidence that probiotics improved growth. In addition, the results from our narrative review on growth outcomes were not cohesive between studies.

For our secondary outcomes, we did observe a significant reduction in the incidence of NEC and all-cause mortality among infants supplemented with probiotics; however, no effect was seen on late-onset sepsis. In addition, non-beneficial results were also seen for infants receiving probiotics. Although not significant, there was some evidence that infants treated with probiotics may have poorer growth at discharge, indicated by lower weight and shorter length. However, this was only a difference of 88 g and 0.3 cm, which might not be clinically relevant. Furthermore, neither the mean weight nor length in the probiotics or control group were considered abnormal for this population. Further, statistically significant evidence of longer hospitalization was seen in the probiotics group. The longer duration of hospital stay exhibited in the probiotics group could potentially be explained by lower weight at discharge, ultimately increasing the need for hospital care before acceptable weight for discharge is attained. All outcomes were graded as low or very low certainty of evidence.

### 4.2. Quality of the Evidence

The Cochrane Risk of Bias Tool 2.0 was used to assess risk of bias for RCTs, and the ROBINS-I tool was applied for non-randomized studies [27,28]. Half of the included RCTs had metalogical weakness, mainly due to adapted study designs, not blinding parents and caregivers, and/or not concealing allocation during analysis. Other problems included limitations in the measurement of outcomes. Further, for some studies, there were concerns with missing data [25,34]. Another concern was the heterogeneity in the population sample size between studies, which increased the risk thfat larger trials and cohorts with significant results had a higher impact on the synthesized results, ultimately increasing the risk of skewing the results. For five of the eleven RCTs, the risk of bias was graded as low [24,29,35,36,38]. For the two non-randomized studies included, the biggest concerns were bias in the measurement of the outcomes and bias due to confounding [23,31]. However, for both studies, all outcomes were in hospital events and data were collected prospectively. The overall risk of bias was graded as moderate.

We used the GRADE approach to assess the certainty of evidence, rating outcomes as either low or very low certainty of evidence.

### 4.3. Agreements and Disagreements with Other Reviews

To our knowledge, this is the first systematic review investigating the effects of probiotic supplements on the primary outcomes of feeding tolerance and growth in solely extremely preterm infants. Although previous systematic reviews have investigated the effect of probiotics, their analyses have focused on NEC, late-onset sepsis and all-cause mortality, whereas feeding tolerance and growth are less studied [15,16,53,54]. Furthermore, earlier RCTs and systematic reviews have predominantly included infants born in gestational week < 37 or very preterm infants < 32 and/or birth weight < 1500 g, rendering their results difficult to interpret for the youngest and most vulnerable subset of this population.

The latest Cochrane review from 2023 on probiotics supplementation in preterm infants assessed the risk of NEC, sepsis and all-cause mortality for extremely preterm infants and extremely low-birth-weight infants in a subgroup analysis [15]. Only data from RCTs were included, and the review did not assess any feeding tolerance or growth outcomes in the younger subgroup. The authors reported a non-statistically significant risk ratio of around 0.9 for NEC, sepsis and all-cause mortality, which is similar to the results of our sensitivity analysis including only RCTs.

A network meta-analysis investigating the effect of multi-strain and single-strain probiotics on preterm infants born below 37 weeks GA found that probiotics reduced feeding intolerance and time to full enteral feeds, but had no effect on weight gain [16]. Another review included infants born below 37 weeks GA from non-randomized studies and found, similar to this review, a lower mean difference in time to full enteral feeds for infants treated with probiotics, but this result was not statistically significant [17]. Furthermore, in a subgroup analysis with extremely preterm infants, a significant risk reduction in NEC was observed, but no effect on late-onset sepsis and all-cause mortality could be observed. Another review including all ages of preterm infants investigated probiotic supplementation on feeding tolerance as the primary outcome, finding a significant improvement in feeding tolerance with lower incidence of feeding intolerance, larger total intestinal feeding time, a larger maximum volume of enteral feeds, and improved weight gain in the probiotics group [18]. However, it is difficult to generalize these results to extremely preterm infants, due to their very different physiology and immaturity.

There are limited numbers of reviews examining the effects of probiotics on growth in extremely preterm infants. Panchal et al. assessed the effect on growth among preterm infants and conducted a subgroup analysis for extremely preterm infants, finding no significant benefits of probiotics on short- or long-term growth [19]. However, they did report a statistically significant effect on short-term growth for the whole cohort of infants born under 37 weeks GA. Another review by Sun et al. included very preterm infants and found no effect on weight gain, similar to our findings [55]. Many other reviews reporting on our primary outcomes have found beneficial effects of probiotic supplementation on feeding tolerance, but limited effects on growth, results which are mirrored in our review.

### 4.4. Implication for Daily Clinical Care

The results from this review highlight a trend towards probiotic supplementation having a slightly beneficial effect in reducing feeding intolerance, but the certainty of evidence was graded as low to very low. Further, probiotics appeared to have limited effects on postnatal growth. The evidence from this review also suggests a risk reduction of NEC and all-cause mortality in infants treated with probiotics, potentially making the prophylactic use of probiotics suitable for clinical use in extremely preterm infants. However, it is important to note that this result was not significant when removing the non-randomized studies, and that these results should be interpreted with caution. This review does not provide any evidence on the ideal type and dosage of probiotics, as the majority of studies used different strain combinations and dosages. The clinical implementation of probiotics should ideally be systematically evaluated, and clinicians should consider their local medical and nutritional guidelines and relevant contexts when considering which product to implement.

### 4.5. Implication for Research

There is limited evidence on the supplementation of probiotics in extremely preterm infants. Previous trials have mainly investigated the effects on extremely preterm infants in subgroup analysis, and trials only including extremely preterm infants have been underpowered. Future RCTs including a sufficient number of extremely preterm infants are warranted. Further, future strain-specific systematic reviews evaluating different types and dosages of probiotics should be conducted. Additionally, trials should not only focus on major neonatal morbidity outcomes, such as NEC and late-onset sepsis, but also strive to report on softer outcomes, such as feeding tolerance and growth, as these outcomes are very common in this population, contribute to overall patient quality of life, and are not extensively researched.

### 4.6. Limitations of This Review

One of the biggest limitations of this review is the heterogenicity among the reported outcomes’ measurements, predominantly for the primary outcomes of feeding tolerance and growth. Due to the diversity of outcome reporting in the included studies, the conducting of meta-analyses and a general presentation of quantitatively summarized results was not possible for all outcomes. Although we only considered NEC Bell Stage ≥II in our analyses, defining NEC is complex and there is a risk that the diagnoses could differ between studies. Furthermore, there was a large variability in the type and dosage of probiotics used alongside the length of the intervention period, making it impossible to draw conclusions on best probiotic-based practices in this population. Further, some studies that we identified as potentially suitable for inclusion did not report a subgroup analysis of extremely preterm infants, and did not get back to us with data, leading us to unfortunately exclude potentially relevant data from this review. Also, the studies included in this review were conducted in various countries and care settings over a large time frame, which may influence the medical treatment received. However, these factors could strengthen the robustness of the results, as the results from this review indicate that probiotics may or may not be effective in different types of settings. Another limitation with this review is the risk of publications bias, especially for the non-randomized trials and smaller clinical trials included in this review, as there is a risk that studies with non-significant results are not published as frequently as those with positive significant outcomes, potentially skewing the synthesized results. However, we attempted to minimize this risk by broadening our search strategy, including studies regardless of publication language. Furthermore, we tried to contact all trials listed as ongoing, recruiting, not yet recruiting or terminated in clincicaltrials.gov to ask if they had published their results or had any unpublished results to share. We also contacted authors with conference abstracts where we could not find a full text publication. However, we did not receive any answers.

## 5. Conclusions

The evidence from this review indicates a trend towards a potential beneficial effect of probiotic supplementation in reducing feeding intolerance, but no to little effect on postnatal growth. A risk reduction of NEC and all-cause mortality in infants treated with probiotics was observed, although the certainty of evidence was low. No guidance can be provided regarding the most beneficial type and dosage of probiotic supplementation. Future studies on the effects of probiotics on feeding and growth outcomes specifically in extremely preterm infants are warranted.

## Figures and Tables

**Table 1 nutrients-17-01228-t001:** Patient demographics and study settings of all included studies.

Study	Country/Setting	Funding/Conflict of Interest	Study Design	Intervention and Comparator	Extremely Preterm Infants (n)	Gestational Age and Body Weight of Extremely Preterm Infants, Mean (SD)
Al-Hosni, 2012 [29]	USA, multicenter	N/A *	Prospective, double-blinded RCT (no placebo)	**Probiotics:** *Lactobacillus rhamnosus GG* and *Bifidobacterium infantis* **Control:** No supplement	**Total:** 101 **Probiotics:** 50 **No probiotics**: 51	**Probiotics** Gestational age: 25.7 (1.4) Birth weight: 778 (138) **No probiotics** Gestational age: 25.7 (1.4) Birth weight: 779 (126)
Alshaikh, 2022 [30]	Canada, single center	N/A	Prospective, open-label RCT (no placebo)	**Probiotics:** *Bifidobacterium breve HA-129, Bifidobacterium bifidum HA-132, Bifidobacterium longum subsp. infantis HA-116, Bifidobacterium longum subsp. longum HA-135* and *Lacticaseibacillus (formerly Lactobacillus) rhamnosus HA-111* **Control:** No supplement	**Total:** 62 **Probiotics:** 31 **No probiotics:** 31	**Probiotics** Gestational age: 25.8 (1.5) **No probiotics** Gestational age: 25.6 (1.3) Birth weight: 751 (132)
Chang, 2022 [31]	Taiwan, single center	Ministry of Science and Technology under the grants MOST 108-2314-B-182-064-MY3, and the Chang Gung Memorial Hospital under the grants NMRPD1J1192.	Prospective cohort study	**Probiotics:** *Lactobacillus acidophilus* and *Bifidobacterium bifidum* **Control:** No supplement	**Total:** 120 **Probiotics:** 70 **No probiotics:** 50	**Probiotics** Gestational age: 26.0 (25.0–27.0) ** Birth weight: 780.0 (689.3–915.0) ** **No probiotics** Gestational age: 26.0 (25.0–27.0) ** Birth weight: 815.0 (757.5–920.0) **
Costeloe, 2016 [32]	England, multicenter	UK National Institute for Health Research Health Technology Assessment programme	Prospective placebo-controlled, double-blinded RCT	**Probiotics:** *B breve BBG-001* Control: Corn starch	**Total:** 634 **Probiotics:** 315 **No probiotics:** 319	Gestational age: N/A Birth weight: N/A
Dobryk, 2023 [33]	Ukraine, multicenter	N/A	Prospective, open-label RCT (no placebo)	**Probiotics: ** *Lactobacillus reuteri * **Control:** No supplement	**Total:** 29 **Probiotics:** 14 **No probiotics:** 15	**Probiotics ** Gestational age: N/A Birth weight: N/A **No probiotics ** Gestational age: N/A Birth weight: N/A
Guney-Varal, 2017 [22]	Turkey, single center	N/A	Prospective RCT (no information about blinding)	**Probiotics:** *Lactobacillus rhamnosus, Lactobacillus casei, Lactobacillus plantorum, Bifidobacterium animalis* **Control:** No supplement	**Total:** 30 * **Probiotics:** 16 **No probiotics:** 14	**Probiotics ** Gestational age: 26 (24–27) ** Birth weight: 978 (248) **No probiotics ** Gestational age: 27 (25–27) ** Birth weight: 1080 (240)
Härtel, 2017 [23]	Germany, multicenter	N/A	Prospective cohort study	**Probiotics:** *Lactobacillus Acidophilus* and *Bifidobacterium infantis* **Control:** No supplement	**Total:** 10,658 **Probiotics:** 7994 **No probiotics:** 2664	**Probiotics** Gestational age: N/A Birth weight: 782 (230) **No probiotics ** Gestational age: N/A Birth weight: 767 (240)
Havranek, 2013 [34]	US, single center	N/A	Prospective, double-blinded RCT (no placebo)	**Probiotics:** *Lactobacillus rhamnosus GG* and *Bifidobacterium infantis* **Control:** No supplement	**Total:** 31 **Probiotics:** 15 **No probiotics:** 16	**Probiotics ** Gestational age: 25.9 (1.3) Birth weight: 856 (105) **No probiotics** Gestational age: 25.9 (1.5) Birth weight: 789 (129)
Jacobs, 2013 [35]	Australia and New Zealand, multicenter	National Health and Research Medical Council of Australia (project grant 454629), The Royal Women’s Hospital Foundation, Melbourne, Australia, and The Angior Family Foundation, Melbourne, Australia	Prospective placebo-controlled, double-blinded RCT	**Probiotics:** *Bifidobacterium infantis, S thermophilus* and *Bifidobacterium lactis* **Control:** placebo (maltodextrin)	**Total:** 454 **Probiotics:** 219 **No probiotics:** 235	**Probiotics ** Gestational age: N/A Birth weight: N/A **No probiotics ** Gestational age: N/A Birth weight: N/A
Manzoni, 2009 [36]	Italy, multicenter	Dicofarm SpA supported this study with a grant and supplied the drugs and placebo used in the study	Prospective placebo-controlled, double-blinded RCT	**Probiotics:** 1. *Bovine lactoferrin* 2. *Bovine lactoferrin + Lactobacillus rhamnosus GG* **Control:** placebo (2 mL of a 5% glucose solution)	**Total:** 114 **Probiotics:** 54 **No probiotics:** 60	**Probiotics ** Gestational age: N/A Birth weight: N/A **No probiotics ** Gestational age: N/A Birth weight: N/A
Patole, 2014 [24]	Australia, single center	Telethon Channel 7 Trust, Western Australia	Prospective placebo-controlled, double-blinded RCT	**Probiotics:** *B. breve M-16V* **Control:** placebo (dextrin)	**Total:** 57 **Probiotics:** 28 **No probiotics:** 29	**Probiotics ** Gestational age: 25.5 (1.3) Birth weight: 742 (191) **No probiotics** Gestational age: 25.9 (1.4) Birth weight: 826 (190)
Totsu, 2014 [25]	Japan, multicenter	No funding source. Study product was provided by Meiji Dairies Corporation, Odawara, Japan	Prospective, placebo-controlled, double-blinded RCT	**Probiotics:** *Bifidobacterium bifidum OLB6378* **Control:** placebo (not specified)	**Total:** 92 **Probiotics:** 48 **No probiotics:** 44	**Probiotics ** Gestational age: 26 (1.1) Birth weight: 838 (190) **No probiotics ** Gestational age: 25 (1.4) Birth weight: 783 (191)
Wejryd, 2019 [38]	Sweden, multicenter	Swedish Research Council (grant number 921.2014-7060), the Swedish Society for Medical Research, the Swedish Society of Medicine, the Research Council for the South-East Sweden, ALF Grants, Region Ostergotland, the Ekhaga Foundation, and BioGaia AB.	Prospective placebo-controlled, double-blinded RCT	**Probiotics:** *L. reuteri DSM 17938* **Control:** placebo (maltodextrin)	**Total:** 134 **Probiotics**: 68 **No probiotics:** 66	**Probiotics ** Gestational age: 25.5 (1.2) Birth weight: 731 (129) **No probiotics** Gestational age: 25.5 (1.3) Birth weight: 740 (148)

* N/A: not available ** Interquartile range

**Table 2 nutrients-17-01228-t002:** Characteristics of ongoing trials.

Corresponding Author, Email	Registration Title	Intervention and Comparator	Outcomes	Registration Date	Trial Status
Benders [39], M.Benders@umcutrecht.nl	NutriBrain: protocol for a randomised, double-blind, controlled trial to evaluate the effects of a nutritional product on brain integrity in preterm infants	**Intervention:** a mixture of probiotics, prebiotics and free amino acid **Control**: maltodextrin, casein and whey protein hydrolysates	Adverse events, growth, number of days with parenteral, days to full enteral nutrition	17 March 2021	Ongoing
Bhadresh [40], bhadreshrvyas@yahoo.co.uk	To study the effect of probiotics preparation Infloran^®^ supplementation in on morbidities in Indian preterm neonates (PrISM)—PrISM	**Intervention:** *Infloran^®^; Lactobacillus Acidophilus, Bifidobacterium Bifidum* **Control:** standard of care	Necrotizing enterocolitis, sepsis, mortality rate, time to full enteral nutrition	30 January 2024	Not Yet Recruiting
Bowornkitiwong [41], walbj@hotmail.com	Effect of probiotics on the incidence of necrotizing enterocolitis in preterm	**Intervention:** *L. acidophilus, L. casei, B. bifidum, B. infantis, B. longum, Lactococcus lactis* **Control:** only breastmilk or formula	Necrotizing enterocolitis, sepsis, mortality rate, time to full enteral nutrition	1 January 2020	Recruiting
Elfving [42], kristina.elfving@gu.se	Probiotic treatment versus placebo to low birth weight neonates for prevention of mortality or undernutrition: two-arm multi-center superiority randomized controlled trial	**Intervention:** *Bifidobacterium * *infantis Bb-02 (DSM 33361) 300 million, Bifidobacterium lactis (BB-12^®^) 350 million,* and *Streptococcus thermophilus (TH-4^®^) 350 million* **Control**: placebo (maltodextrin)	All-cause mortality, sepsis, growth	26 April 2023	Pending
Infant Bacterial Therapeutics AB (IBT) [43], clinical@ibtherapeutics.com	Study on the prevention of necrotizing enterocolitis (a severe inflammation and death go intestines) in premature infants	**Intervention:** *Lactobaciullus reuteri* **Control:** N/A *	Necrotizing enterocolitis, all-cause mortality, duration of hospitalization, growth and feeding tolerance	21 March 2019	Authorized recruitment may be ongoing or finished
Masood [44], drimranmasood@iub.edu.pk	Evaluation of the clinical and growth-related significance of probiotics in preterm infants	**Intervention:** 1: *Lactobacillus rhamnosus* 2: *Bifidobacterium BB-12, Lactobacillus paracasei, L casei-431, Streptococcus thermophilus TH-4 (Amybact)* **Control:** Dextrose water 10%	Morbidity (general), growth, feeding patterns	4 May 2023	Recruiting
Mukthar [45], drmushtaq816@gmail.com	Role of prophylactic microbial supplements in prevention of blood stream infection and intestinal tract injury in premature neonates	**Intervention:** Mixture of *lactobacillus acidophilus, lactobacillus rhamnosus, bifidobacterium longum* and *streptomyces boulardii* **Control:** only breastmilk/formula	Necrotizing enterocolitis, hospital stay, time to establish full feeds, episodes of feeding intolerance, nosocomial sepsis, all-cause mortality	20 April 2018	Open to recruitment
Pandey [46], sshahdoc@gmail.com	Lactobacillus Rhamnosus GG to reduce NEC, sepsis and mortality in VLBW infants –A Randomised Controlled Trial	**Interevention:** *Lactobacillus Rhamnosus GG* (liquid form) **Control:** only mothers breastmilk or donor milk	Necrotizing enterocolitis, sepsis, all-cause mortality, time to full feeds, duration of parental nutrition, hospital stay, growth	4 March 2021	Closed to recruitment
Rakow [47], alexander.rakow@regionstockholm.se	Probiotic Supplementation in Extremely Preterm Infants in Scandinavia (PEPS)	**Intervention:** *Bifidobacterium infantis Bb-02 (DSM 33361) 300 million, Bifidobacterium lactis (BB-12^®^) 350 million, and Streptococcus thermophilus (TH-4^®^) 350 million* **Control:** placebo (maltodextrin)	Necrotizing enterocolitis, sepsis, all-cause mortality, time to full feeds, duration of parenteral nutrition, hospital stay, use of antibiotics, growth, feeding tolerance	16 October 2023	Recruiting
Taslimi Taleghani [48], naeemetaslimi@yahoo.com	The effect of early oral probiotics prescription on feeding intolerance, regain birth weight and secondary outcomes in very low birth weight	**Intervention:** *Bifidobacterium lactis, Bifidobacterium infantis* and *Streptococcus thermophilus 3* **Control:** placebo	Feeding tolerance, sepsis, necrotizing enterocolitis, time to full enteral nutrition, hospital stay, days to attain birth weight	4 August 2022	Recruiting
Tunpowpong [49], tuinoi1@gmail.com	Randomized controlled trial comparing gut microbiota of preterm neonates receiving Infloran versus not receiving Infloran	**Intervention:** *B. bifidum* and *L. acidophilus * **Control:** standard feeding protocol	Necrotizing enterocolitis, gut microbiome alteration	28 September 2022	Recruiting
van Wyk [50], lizelle@sun.ac.za	The role of a multi-strain probiotic in very low birth weight infants.	**Intervention:** *Labinic probiotic* **Control:** placebo	Rectal colonization, postnatal growth, bloodstream infections, incidence of necrotizing enterocolitis and severity, time to full feeds	4 April 2020	Pending
Vaswani [51], ndvaswani@hotmail.com	To study the effect of supplementation of probiotics on enteral feed tolerance in very low birth weight neonates—Randomized Control Trial	**Intervention:** N/A **Control:** formula milk as placebo	Time to full enteral nutrition, feeding tolerance, necrotizing enterocolitis, sepsis, length of hospital stay	4 October 2023	Not Yet Recruiting
Wang [52], 568241134@qq.com	Clinical study on the effect of probiotics on intestinal metabolites and serum inflammatory factors on the outcome of premature infants	**Intervention:** probiotics (not specified) **Control**: Dextrose 5%	Age at return to birth weight, time to central cannulation, weight gain (g/day), length of hospitalization, late onset sepsis, incidence of feeding intolerance, necrotizing enterocolitis	5 January 2024	Completed, no publication

* N/A: not available.

**Table 3 nutrients-17-01228-t003:** Characteristics of studies awaiting classification.

Corresponding Author, Email	Title	Intervention and Comparator	Outcomes	Publication Date	Reason for Awaiting Classification
Singh [37], balpreet-singh@iwk.nshealth.ca	Probiotics for preterm infants: A National Retrospective Cohort Study	**Intervention: ** 1. *Bifidobacterium species (B. breve, B. bifidum, B. infantis, and B. longum)* and Lactobacillus rhamnosus GG 2. *Lactobacillus reuteri* **Control: ** no probiotics	Days with parenteral nutrition, mortality, NEC, late-onset sepsis, length of hospital stay	28 January 2019	Study included babies < 29 gestational weeks. Awaiting data for babies born <28 gestational weeks

**Table 4 nutrients-17-01228-t004:** Primary outcome measurements as presented by the original studies, results for primary and secondary outcomes relevant for this systematic review and risk of bias.

Study	Outcomes (as Mentioned by Study Authors for Infants < 28 weeks GA *)	Primary Outcomes for This Systematic Review (Feeding Tolerance, Growth)	Secondary Outcomes for This Systematic Review (Death, NEC, Sepsis, Hospitalization, ABs)	Risk of Bias
Al-Hosni, 2012[29]	**Primary ** Feeding volume Growth velocity Weight gain per day Weight < 10 percentile at 34 GW **Secondary ** Antimicrobial days Antibacterial days Antifungal days Death at 34 GW NEC ** Bell Stage I–III NEC surgery Sepsis (bacterial and fungal) Weight 34 GW	**Feeding volume (average mL/kg/day) first 28 days ** Intervention: 59 Control: 71 **Growth velocity (g/kg/day) ** Intervention: 14.9 Control: 12.6 **Weight < 10 percentile at 34 GW ** Intervention: 27/47 Control: 28/47 **Weight at 34 GW (grams) ** Intervention: 1656 Control: 1599	**Death ** Intervention: 3/50 Control: 4/51 **NEC Bell Stage II-III ** Intervention: 2/50 Control: 2/51 **Sepsis (bacterial and fungal) ** Intervention: 13/50 Control: 16/51	Low
Alshaikh, 2022 [30]	**Primary ** Changes in fecal microbiota **Secondary ** Duration parenteral nutrition Enteral nutrition interrupted Change in weight z-score between birth and 40 GW Growth velocity day 7–36 GW Head circumference at 40 GW Head circumference Z-score at 40 GW Hospitalization Late-onset sepsis Length at 40 GW Length Z-score at 40 GW NEC Bell Stage IIIII Time to full enteral nutrition Weight at 40 GW Weigth Z-score at 40 GW Weight < 10 percentile at 40 GW	**Days full enteral nutrition (120 mL/kg/day) ** Intervention: 12 Control: 13 **Duration parenteral nutrition (days) ** Intervention: 20 Control: 17 **Enteral nutrition interrupted (>24 h), episodes (n) ** Intervention: 3 Control: 4 **Change in weight z-score between birth and 40 GW** Intervention: −1.2 Control: −1.2 **Growth velocity day 7–36 GW (g/kg/day) ** Intervention: 17.4 Control: 18.5 **Head circumference at 40 GW (cm) ** Intervention: 32.4 Control: 33 **Head circumference Z-score at 40 GW ** Intervention: −1.8 Control: −1.4 **Length at 40 GW (cm) ** Intervention: 48 Control: 48.1 **Length Z-score at 40 GW ** Intervention: −2.5 Control: −2.5 **Weight at 40 GW (grams) ** Intervention: 3484 Control: 3516 **Weight Z-score at 40 GW ** Intervention: −1.7 Control: −1.6 **Weight < 10 percentile at 40 GW ** Intervention: 15/31 Control: 17/31	**Hospitalization (median days) ** Intervention: 107.5 Control: 96 **NEC ** Intervention: 2/31 Control: 0/31 **Late-onset sepsis ** Intervention: 8/31 Control: 3/31	Moderate
Chang, 2022 [31]	**Primary ** Fecal microbiota **Secondary ** Duration parenteral nutrition Hospitalization In-hospital mortality NEC Bell Stage II-III Late-onset sepsis	**Duration of parenteral nutrition (days) ** Intervention: 29 Control: 35.5	**Hospitalization (median days) ** Intervention: 96.5 Control: 98 **In-hospital mortality ** Intervention: 3/70 Control: 3/50 **NEC Bell Stage II-III ** Intervention: 5/70 Control: 2/50 **Late-onset sepsis ** Intervention: 33/70 Control: 35/50	Serious
Costeloe, 2016 [32]	**Primary ** Culture-proven sepsis > 72 h after birth In-hospital mortality NEC Bell Stage II-III	N/A ***	**Culture-proven sepsis ** Intervention: 63/315 Control: 58/319 **In-hospital mortality ** Intervention: 42/315 Control: 53/319 **NEC Bell Stage II-III ** Intervention: 48/315 Control: 52/319	Serious
Dobryk, 2023 [33]	**Primary ** NEC Bell Stage II-III **Secondary** Antibiotic use Days to full enteral feeds Hospitalization Late-onset sepsis Mortality Reduced feeding tolerance Weight at 36 GW	**Days to full enteral feeds (160 mL/kg/day) ** Intervention: 37 Control: 40 **Reduced feeding tolerance (mean number of episodes) ** Intervention: 1 Control: 3 **Weight at 36 GW (grams) ** Intervention: 2190 Control: 2160	**Hospitalization (mean days) ** Intervention: 96 Control: 101 **Late-onset sepsis ** Intervention: 7/14 Control: 7/15 **Mortality ** Intervention: 1/14 Control: 2/15 **NEC Bell Stage II-III ** Intervention: 2/15 Control: 1/14	Moderate
Guney-Varal, 2017 [22]	**Primary ** NEC Bell Stage II-III **Secondary ** Days to 100 mL/kg/day Days to 150 mL/kg/day Duration parenteral nutrition Feeding intolerance episodes ≥ 3 times Hospitalization Mortality Late-onset sepsis	**Days to 100 mL/kg/day ** Intervention: 13 Control: 22 **Days to 150 mL/kg/day ** Intervention: 16 Control: 27 **Duration parenteral nutrition (mean days) ** Intervention: 16 Control: 29 **Feeding intolerance episodes ≥3 times (defined as abdominal distension, gastric residue or vomiting) ** Intervention: 9 Control: 14	**Hospitalization (mean days) ** Intervention: 58 Control: 56 **Mortality ** Intervention: 1/16 Control: 7/14 **NEC Bell Stage II-III ** Intervention: 0/16 Control: 2/14 **Late-onset sepsis ** Intervention: 5/16 Control: 4/14	Moderate
Härtel, 2017 [23]	**Primary ** Growth velocity (grams/day) Head circumference at discharge Length at discharge Weight at discharge **Secondary** Hospitalization In hospital mortality NEC Bell Stage II-III Late-onset sepsis	**Growth velocity (grams/day) ** Intervention: 21 Control: 20 **Head circumference at discharge (cm) ** Intervention: 32.6 Control: 32.6 **Length at discharge (cm) ** Intervention: 46.2 Control: 44.6 **Weight at discharge (grams) ** Intervention: 2526 Control: 2741	**Hospitalization (mean days) ** Intervention: 94 Control: 89 **In hospital mortality ** Intervention: 405/7994 Control: 360/2663 NEC Bell Stage II-III Intervention: 350/7993 Control: 155/2664 **Late-onset sepsis ** Intervention: 1439/7994 Control: 515/2663	Moderate
Havranek, 2013 [34]	**Primary ** Blood flow velocity **Secondary ** Days to full enteral nutrition	**Time to full enteral nutrition (150 mL/kg/day) (days) ** Intervention: 23.9 Control: 22.1	N/A Sub-study of Al-Hosni. Primary and secondary outcomes are already presented and are not reported again.	Moderate
Jacobs, 2013 [35]	**Primary ** Late-onset sepsis **Secondary ** NEC Bell Stage II-III	N/A	**NEC Bell Stage II-III ** Intervention: 11/219 Control: 17/235 **Late-onset sepsis ** Intervention: 54/219 Control: 55/235	Low
Manzoni, 2009 [36]	**Primary ** First onset of late-onset sepsis **Secondary** Late-onset sepsis	N/A	**Late-onset sepsis (bacterial and fungal) ** Intervention: 6/37 Control: 19/46	Low
Patole, 2014 [24]	**Primary ** Fecal microbiota **Secondary ** Duration parenteral nutrition Growth retardation at discharge Head circumference at discharge Hospitalization Mortality at discharge NEC Bell Stage II-III Late-onset sepsis Time to 50 mL/kg/day enteral nutrition Time to 100 mL/kg/day enteral nutrition Time to 150 mL/kg/day enteral nutrition Weight at discharge Length at discharge	**Duration parenteral nutrition (mean days) ** Intervention: 20 Control: 17 **Growth retardation at discharge (no clear definition) ** Intervention: 19/28 Control:10/28 **Head circumference at discharge (mean cm) ** Intervention: 34 Control: 34 **Length at discharge (mean cm) ** Intervention: 48/18 Control: 51/14 **Time to 50 mL/kg/day enteral nutrition (days) ** Intervention: 14 Control: 13 **Time to 100 mL/kg/day enteral nutrition (days) ** Intervention: 18 Control: 16 **Time to 150 mL/kg/day enteral nutrition (days) ** Intervention: 29 Control: 23 **Weight at discharge (mean grams) ** Intervention: 2933 Control: 3173	**Hospitalization (mean days) ** Intervention: 105 Control: 98 **Late-onset sepsis ** Intervention: 12/28 Control: 7/28 **Mortality at discharge ** Intervention: 0/28 Control: 0/28 **NEC Bell Stage II-III ** Intervention: 0/28 Control: 0/28	Low
Totsu, 2014 [25]	**Primary ** Established enteral nutrition (>100 mL/kg/day) **Secondary ** Head circumference at discharge Late-onset sepsis **** Mortality at discharge NEC Bell Stage II-III Weight at discharge	**Time to 100 mL/kg/day enteral nutrition (days) ** Intervention: 13 Control: 13 **Head circumference at discharge (mean cm) ** Intervention: 35/48 Control: 35/40 **Weight at discharge (mean grams) ** Intervention: 2926/48 Control: 3008/48	**Late-onset sepsis **** ** Intervention: 6/48 Control: 9/44 **Mortality at discharge ** Intervention: 0/48 Control: 0/44 **NEC Bell Stage II-III ** Intervention: 0/48 Control: 0/44	Moderate
Wejryd, 2019 [38]	**Primary ** Time to full enteral nutrition **Secondary ** Culture-proven sepsis Duration parenteral nutrition Gastric residuals Head circumference day 14 (cm) Head circumference day 28 (cm) Head circumference 36 GW (cm) Interrupted feeding Length day 14 (cm) Length day 28 (cm) Length 36 GW (cm) Mortality NEC Bell Stage II-III Stools per week Weight day 7 (g) Weight day 14 (g) Weight day 21 (g) Weight day 28 (g) Weight 36 GW (g)	**Duration parenteral nutrition (mean days) ** Intervention: 24.1 Control: 23.1 **Gastric residuals (larger than 2 mL/kg and exceed volume of previous meal, mean n) ** Intervention: 3 Control: 3.8 **Head circumference day 14 (mean cm) ** Intervention: 23.2 Control: 23.5 **Head circumference day 28 (mean cm) ** Intervention: 25.2 Control: 24.9 **Head circumference 36 GW (mean cm) ** Intervention: 31.2 Control: 31 **Interrupted feeding (mean days) ** Intervention: 5.5 Control: 6 **Length day 14 (mean cm) ** Intervention: 33.9 Control: 34.3 **Length day 28 (mean cm) ** Intervention: 35.9 Control: 35.5 **Length 36 GW (mean cm) ** Intervention: 43 Control: 43.4 **Time to full enteral nutrition 150 mL/kg/day (median days) ** Intervention: 15 Control: 15 **Weight day 7 (mean grams) ** Intervention: 767 Control: 761 **Weight day 14 (mean grams) ** Intervention: 867 Control: 889 **Weight day 21 (mean grams) ** Intervention: 975 Control: 972 **Weight day 28 (mean grams) ** Intervention: 1075 Control: 1074 **Weight 36 GW (mean grams) ** Intervention: 2303 Control: 2348	**Culture-proven sepsis ** Intervention: 25/68 Control: 23/66 **Mortality ** Intervention: 5/68 Control: 5/66 **NEC Bell Stage II-III ** Intervention: 7/68 Control: 8/66	Low

* GA: Gestational age. ** NEC: necrotizing enterocolitis. *** N/A: Not avaliable **** Defined as sepsis ≥ 1 week after birth.

**Table 5 nutrients-17-01228-t005:** Summary of findings table including main results and certainty of evidence.

Outcomes	Anticipated Effect		Mean Difference/Risk Ratio (95% CI)	Number of Participants	Certainty of Evidence (GRADE)
	Risk with Control Group	Risk with Probiotics Group			
**Feeding intolerance**					
Days to full enteral nutrition (150 mL/kg/day)	21.8	21	1.1 days lower (7.83 lower to 5.56 higher)	251	Very low
Duration of parenteral nutrition (days)	24.3	22	2.4 days lower (7.44 lower to 2.58 higher)	402	Very low
**Growth**					
Weight at discharge (grams)	2824	2736	88 g lower (205 lower to 30 higher)	10,988	Very low
Length at discharge (cm)	46.8	47.1	0.3 cm higher (1.1 lower to 1.8 higher)	11,068	Very low
Head circumference at discharge (cm)	33.2	33	0.02 cm lower (0.12 lower to 0.09 higher)	10,876	Low
**Necrotizing enterocolitis**	7 per 100	5 per 100	RR 0.80 (0.68 to 0.93)	12,369	Low
**All-cause mortality** (during hospital stay)	13 per 100	7 per 100	RR 0.56 (0.33 to 0.93)	11,853	Very low
**Late-onset sepsis**	21 per 100	20 per 100	RR 0.95 (0.79 to 1.11)	12,452	Very low
**Hospitalization (days)**	88	93	5 days higher (3.7 higher to 6.6 higher)	10,887	Very low

## Data Availability

The original contributions presented in this study are included in the article/Appendix A. Further inquiries can be directed to the corresponding authors.

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
