# Peer review of "Probiotic Supplements Effect on Feeding Tolerance, Growth and Neonatal Morbidity in Extremely Preterm Infants: A Systematic Review and Meta-Analysis"

_nutrients, 2025, doi:10.3390/nu17071228_

Round 1
Reviewer 1 Report
Comments and Suggestions for Authors
The manuscript is very well written and targets an important topic, i.e. evidence of probiotics to improve feeding intorlenace in üpreterm infants < 28 wk of gestation and other clinically relevant outcomes. Introduction is sound, methods for meta-analysis are sound, the results are well described and the discussion is reflecting the current literature and addresses limitations adequately.There are only a few minor suggestions:
1) please state in the abstract your eligibility criteria: e.g. "We limited our population to extremely preterm infants, defined as those born before 28 weeks GA.."
2) The authors state that six studies reported on weight at discharge and included a total of 10,988 extremely preterm infants [23-25, 28, 31, 36]. The mean weight at discharge was 88 grams lower in the probiotics group, however not statistically significant (95% CI, 205 lower to 30 higher) and the certainty of evidence was very low.
This meta-analysis includes observational cohort studies such as Härtel (Scientif Rep 2017) which includes registry data from > 40 NICUs. As thee is no unirform guideline across the neonatal network, most units would use probiotics in those babies who are most vulnerable, < 1000g, < 27 weeks which have other aspects related to reduced weight gain (increased work of breathing, inflammation ...). This is a selection bias of most vulnerable in the experimental group and certainly not a causal link to probiotic treatment. The authors may consider to clarify that.
3) In the discussion the authors may consider to discuss the complexity of defining intestinal injury/NEC which explains some of the great heterogeneity of studies. For future study endpoints, the scientific community needs to find consensus including biomarkers/AI, as reflected by recent work of Joe Neu and colleagues.
Author Response
Reviewer #1: The manuscript is very well written and targets an important topic, i.e. evidence of probiotics to improve feeding intorlenace in üpreterm infants < 28 wk of gestation and other clinically relevant outcomes. Introduction is sound, methods for meta-analysis are sound, the results are well described and the discussion is reflecting the current literature and addresses limitations adequately. There are only a few minor suggestions:
- 1) please state in the abstract your eligibility criteria: e.g. "We limited our population to extremely preterm infants, defined as those born before 28 weeks GA.."
(Reviewer #1, comment #1)
Great suggestion, we have added this to the abstract.
- 2) The authors state that six studies reported on weight at discharge and included a total of 10,988 extremely preterm infants [23-25, 28, 31, 36]. The mean weight at discharge was 88 grams lower in the probiotics group, however not statistically significant (95% CI, 205 lower to 30 higher) and the certainty of evidence was very low.
This meta-analysis includes observational cohort studies such as Härtel (Scientif Rep 2017) which includes registry data from > 40 NICUs. As thee is no uniform guideline across the neonatal network, most units would use probiotics in those babies who are most vulnerable, < 1000g, < 27 weeks which have other aspects related to reduced weight gain (increased work of breathing, inflammation ...). This is a selection bias of most vulnerable in the experimental group and certainly not a causal link to probiotic treatment. The authors may consider to clarify that.
(Reviewer #1, comment #2)
Thank you for your insightful comment. As for all included studies, from the study by Härtel et al we only extracted data for infants born extremely preterm and thus this alone cannot explain the lower weight gain in the probiotics group. Unfortunately, we did not have data on the mean gestational age, and whether that differed between probiotics and control group. Although, the birth weight was similar. There was however a significantly higher number of infants receiving probiotic (n=7994) than those who did not (n=2664). As you describe, with the limited uniform guidelines on when to provide probiotics and possible favoring probiotic supplementation to the sickest infants, this could potentially explain the lower weight at discharge in the probiotics group. However, as we did not collected data on other comorbidity such as respiratory work and lung function, we do not wish to speculate whether there was a difference between probiotics and control group in this population.
- 3) In the discussion the authors may consider to discuss the complexity of defining intestinal injury/NEC which explains some of the great heterogeneity of studies. For future study endpoints, the scientific community needs to find consensus including biomarkers/AI, as reflected by recent work of Joe Neu and colleagues.
(Reviewer #1, comment #3)
We completely agree with the reviewer that defining NEC is complex and could contribute to the heterogeneity between studies. In the discussion section under “4.6 Limitations of this review” line 648-650, we discuss that even if this review only considered NEC Bell Stage ≥II, there is a risk that diagnosis could differ between studies. However, we have tried to elaborate on this further.
Reviewer 2 Report
Comments and Suggestions for Authors
Thank you for allowing me to review this manuscript. This topic is of extreme importance to evaluate the available literature on the impact of probiotics, especially in light of recent changes to probiotic use based on recommendations by governing bodies and discontinuation of available probiotic products. My comments and suggestions for this manuscript are as follows:
ï‚§ Line 42-43: Suggest modifying “Feeding intolerance leads to delay or interruptions of enteral feeds, causing prolonged parenteral nutrition” to “Feeding intolerance leads to a delay or interruptions in providing full enteral nutrition, resulting in prolonged need for parenteral nutrition provision”.
ï‚§ Line 45: “gastric residuals” can be present but not be due to feeding intolerance. Therefore, perhaps this should be rephrased to “high gastric residuals”, though notably, checking gastric residuals is not recommended in this population.
ï‚§ Line 45-47: I suggest rephrasing this statement. There is a lot implied here. Increased apneic and bradycardic episodes are not always associated with “feeding intolerance”. It requires careful clinical evaluation and evaluating the entire big picture of the infant’s clinical progress. Likewise, poor intestinal motility and delayed stool passage can contribute to “feeding intolerance”.
o I likewise suggest a closer evaluation and rephrasing of the rest of this paragraph (through Line 57). I think it is important to note that clinical nutrition management practices vary drastically among clinicians and units. Further, it cannot be assumed that because an infant may have feeding intolerance that their growth will suffer (this can be avoided if adequate parenteral nutrition is provided). I also suggest taking out terms like “___will cause…” and rephrase to something like “___ may contribute to…” (to remove assumptions).
o Somewhere in this manuscript, it must also be clearly acknowledged that terms like “feeding intolerance” or “full enteral feedings” also vary between clinicians and units.
ï‚§ Line 61: Please rephrase the start of this sentence for grammatical improvement.
ï‚§ Line 59-67: More references are needed here.
ï‚§ Within the Methods, I think there are many variables included that are not well defined. This is appropriately mentioned later in the manuscript, but I suggest adding something in this section that definitions or criteria for each variable may have varied between studies.
ï‚§ In Table 1, can you please explain how gestational age is “N/A” for selected studies when gestational age is an inclusion variable for this study?
ï‚§ One thing to consider regarding growth at discharge is that infants may have been discharged home at differing days of life and/or gestational ages.
ï‚§ Given decreasing gestational age of infants receiving care, is there a lowest limit for infants who received probiotics in these studies (e.g. 22, 23, 24 weeks, etc.)? If so, I think this should be clearly stated for further context.
ï‚§ As previously stated, this is an important manuscript. Given the wide range of variables evaluated, I suggest splitting this into two manuscript. I suggest one focus on more concrete outcomes like NEC, mortality, hospitalization duration, sepsis. The second manuscript can focus on less concrete outcomes (with variable definitions) like feeding tolerance, time to full feedings, growth, etc.
Author Response
Reviewer #2: Thank you for allowing me to review this manuscript. This topic is of extreme importance to evaluate the available literature on the impact of probiotics, especially in light of recent changes to probiotic use based on recommendations by governing bodies and discontinuation of available probiotic products. My comments and suggestions for this manuscript are as follows:
- Line 42-43: Suggest modifying “Feeding intolerance leads to delay or interruptions of enteral feeds, causing prolonged parenteral nutrition” to “Feeding intolerance leads to a delay or interruptions in providing full enteral nutrition, resulting in prolonged need for parenteral nutrition provision”.
(Reviewer #2, comment #1)
Thank you for your enthusiasm on the topic and for your suggestion. We have made some alterations to the sentence. However, as feeding intolerance can be defined as interruptions or delay in increasing enteral feeds even before full enteral nutrition has been established, we have decided not to alter that part.
- Line 45: “gastric residuals” can be present but not be due to feeding intolerance. Therefore, perhaps this should be rephrased to “high gastric residuals”, though notably, checking gastric residuals is not recommended in this population.
(Reviewer #2, comment #2)
We agree with the reviewer that checking gastric residuals is not generally recommended in this population. We also recognize that gastric residuals may be due to different factors and not dependent on feeding intolerance. Accordingly, we have added a ”can” in the sentences to emphasize that this can be a sign of feeding intolerance but not a necessity. In addition, this definition is also used as a variable for feeding intolerance in some of the included studies, which is why we deem it as appropriate to introduce the term in the background section.
- Line 45-47: I suggest rephrasing this statement. There is a lot implied here. Increased apneic and bradycardic episodes are not always associated with “feeding intolerance”. It requires careful clinical evaluation and evaluating the entire big picture of the infant’s clinical progress. Likewise, poor intestinal motility and delayed stool passage can contribute to “feeding intolerance”.
(Reviewer #2, comment #3)
Thank you for pointing this out, we fully agree with the statement and have added poor intestinal motility and delayed stool passage as contributors to feeding intolerance. We are not suggesting that apnea and bradycardia alone are always are associated with feeding intolerance, however in combination with gastric residuals, vomiting, and abdominal distension can be a sign of feeding intolerance and in worst case scenario NEC or late-onset sepsis.
- I likewise suggest a closer evaluation and rephrasing of the rest of this paragraph (through Line 57). I think it is important to note that clinical nutrition management practices vary drastically among clinicians and units. Further, it cannot be assumed that because an infant may have feeding intolerance that their growth will suffer (this can be avoided if adequate parenteral nutrition is provided). I also suggest taking out terms like “___will cause…” and rephrase to something like “___ may contribute to…” (to remove assumptions).
(Reviewer #2, comment #4)
Excellent suggestion. We have added a sentence highlighting the difference in clinical management of feeding tolerance (line 56).
We are not sure which sentence the reviewer is referring to with the term “will cause”. We have reviewed the whole manuscript carefully to make sure that such assumptions are not made.
- Somewhere in this manuscript, it must also be clearly acknowledged that terms like “feeding intolerance” or “full enteral feedings” also vary between clinicians and units.
(Reviewer #2, comment #5)
Absolutely, in the method, results and discussion section, we emphasize the
heterogeneity in the reported outcomes especially for feeding tolerance and growth outcomes. In addition, we have also added a sentence in the background section (line 45-46), describing that the definition of feeding tolerance is not reported unitary in research or clinical practice.
- Line 61: Please rephrase the start of this sentence for grammatical improvement.
(Reviewer #2, comment #6)
Thank you for noticing. We have corrected the sentences.
- Line 59-67: More references are needed here.
(Reviewer #2, comment #7)
Another reference has been added in this section. We are happy to accept suggestions on potentially relevant papers that we have missed.
- Within the Methods, I think there are many variables included that are not well defined. This is appropriately mentioned later in the manuscript, but I suggest adding something in this section that definitions or criteria for each variable may have varied between studies.
(Reviewer #2, comment #8)
We recognize that some of the outcome variables under “2.6 Data items” where not appropriately described which has now been modified. However, following the PRSIMA guidelines for reporting on systematic reviews, we believe that it is more appropriate to present differences in outcome variables between studies in the result section, primarily in Table 4.
- In Table 1, can you please explain how gestational age is “N/A” for selected studies when gestational age is an inclusion variable for this study?
(Reviewer #2, comment #9)
Certainly, in the last column of Table 1, we present the mean gestational age and birth weight. However, some of the included studies reported on extremely preterm infants but did not specify the mean gestational age and/or birth weight for those infants. For those studies, that column have been marked with N/A.
- One thing to consider regarding growth at discharge is that infants may have been discharged home at differing days of life and/or gestational ages.
(Reviewer #2, comment #10)
Yes, we agree that this would have been very interesting. Unfortunately, we were not able to extract this information from the included studies.
- Given decreasing gestational age of infants receiving care, is there a lowest limit for infants who received probiotics in these studies (e.g. 22, 23, 24 weeks, etc.)? If so, I think this should be clearly stated for further context.
(Reviewer #2, comment #11)
Very relevant suggestion. We agree that this is important for future context and clinical care. As previously described, the mean gestational for included infants was not available for all studies. In addition, in studies including information about mean gestational age, the authors generally presented the standard deviation or interquartile range but not the lowest limit of gestational age for included infants. However, the range of gestational age was reported in five of the included studies with Totsu et al including the infants with lowest gestational age at birth (week 22). We have now included information about this in our manuscript under section 3.3 Study population.
- As previously stated, this is an important manuscript. Given the wide range of variables evaluated, I suggest splitting this into two manuscript. I suggest one focus on more concrete outcomes like NEC, mortality, hospitalization duration, sepsis. The second manuscript can focus on less concrete outcomes (with variable definitions) like feeding tolerance, time to full feedings, growth, etc.
(Reviewer #2, comment #12)
We understand your suggestion to split the manuscript into two parts. However, we believe that keeping it as a single manuscript is essential, as all the evaluated variables are closely interconnected within the clinical context of extremely preterm infants.

Reviewer 3 Report
Comments and Suggestions for Authors
This is a systemic review of the probiotic effect on the mortality and morbidity of extremely premature infants. Primary outcomes are “feeding intolerance” and “growth during a defined age”. One major weakness of the study is the very low response rate (4/29) from the authors. This, unfortunately, significantly decreases the validity of the meta-analysis. Fourteen (14) studies were included in the meta-analysis according to the standard and well-accepted criteria. There is a significant decrease in necrotizing enterocolitis (NEC) and all-cause mortality in the probiotic-treated infants. The results are exciting, but the certainty of evidence is graded as low or very low after including 14,888 subjects. This strongly suggests if there is a benefit then the number-need-to-treat (NNT) will be very large.
I am confused about how many studies were included. 29 were mentioned in the “Materials and Methods” while 28 were described in “Results”. I cannot find an explanation for the discrepancy. Am I missing something? Fourteen studies were included in the meta-analysis as described in abstract but 28 were mentioned in the materials and methods. I recommend you providing consistent information. It is foreseeable that such meta-analysis is difficult to conduct as different kind of probiotics (single train versus multiple strain, amount of the spores), when probiotics were started, single center versus multi-site, feeding policy/routine, etc. all confound the findings.
In results, large amount of ambivalent or contradictory findings are described with low certainty. Significant results only occur in secondary outcomes (NEC and all-cause mortality) with significantly longer hospitalization. The meta-analysis is straightforward if you follow the guidelines. I am curious the importance of your subgroup analysis which in fact takes away some of your positive findings. You did not attempt to address about the longer hospitalization which caught my attention. Was there a reason why not? I do not encourage you to address too much about those potentially beneficial but statistically insignificant effect. The explanation that longer hospitalization in probiotic group might be due to the lower discharge weight does not make too much of sense to me. As there was no difference in the initial birth weight and GA, the your explanation might indicate that probiotics slows down the weight gain.
Some abbreviations, such as Nct, Ctri, and ChiCtr, need clarification. They appear both in the main text and supplementary material without providing any information.
Author Response
- Reviewer #3: This is a systemic review of the probiotic effect on the mortality and morbidity of extremely premature infants. Primary outcomes are “feeding intolerance” and “growth during a defined age”. One major weakness of the study is the very low response rate (4/29) from the authors. This, unfortunately, significantly decreases the validity of the meta-analysis. Fourteen (14) studies were included in the meta-analysis according to the standard and well-accepted criteria. There is a significant decrease in necrotizing enterocolitis (NEC) and all-cause mortality in the probiotic-treated infants. The results are exciting, but the certainty of evidence is graded as low or very low after including 14,888 subjects. This strongly suggests if there is a benefit then the number-need-to-treat (NNT) will be very large.
(Reviewer #3, comment #1)
Certainly, when reviewing subgroups in systematic reviews there is unfortunately a risk of not being able to include all studies due to missing data. However, we want to emphasize that rather than to only include studies with the correct data in the original article, attempting to obtain data from all potential studies is the best course of action and recommended by the PRISMA Guidelines and Cochrane. The response rate 4/29 where not from our 28 included studies. Please see our response under comment #2.
- I am confused about how many studies were included. 29 were mentioned in the “Materials and Methods” while 28 were described in “Results”. I cannot find an explanation for the discrepancy. Am I missing something? Fourteen studies were included in the meta-analysis as described in abstract but 28 were mentioned in the materials and methods. I recommend you providing consistent information. It is foreseeable that such meta-analysis is difficult to conduct as different kind of probiotics (single train versus multiple strain, amount of the spores), when probiotics were started, single center versus multi-site, feeding policy/routine, etc. all confound the findings.
(Reviewer #3, comment #2)
We apologize for the confusion. We believe that the reviewer is referring to the missing data described in Methods section “2.7 Missing data”, where we describe that we requested data from studies that could potentially be included in our systematic review. These 29 studies are not the same as the 28 studies included in our final population. For example, in studies included babies with birth weight >1000 grams, we contacted the authors to try to receive information about potential infants born <28 weeks GA. However, we only received data from four of them which were later included in our systematic review.
- In results, large amount of ambivalent or contradictory findings are described with low certainty. Significant results only occur in secondary outcomes (NEC and all-cause mortality) with significantly longer hospitalization. The meta-analysis is straightforward if you follow the guidelines. I am curious the importance of your subgroup analysis which in fact takes away some of your positive findings.
(Reviewer #3, comment #3)
Thank you for your astute comment. Differences between the main results and those observed in subgroup or sensitivity analyses can arise due to many reasons such as variations in study populations and methodological differences namely use of different probiotics. While these analyses may attenuate some positive findings, they also provide a more nuanced understanding of the data and help prevent overinterpretation of results driven by specific subsets of studies. The subgroup analysis consisted of testing RCTs and non-randomized studies separately. A lower risk ratio for NEC and all-cause mortality as well as a higher mean difference in hospitalization for the probiotics group, was still observed, however no longer statistically significant for any of these outcomes. We believe that one major factor was that the study by Härtel et al was a large cohort study weighing heavily due to its large sample size ultimately increasing the confidence interval as this study was removed from analysis. If anything, this emphasizes the immense need for a sufficiently powered randomized trial among extremely preterm infants which we acknowledge in our discussion and conclusions.
- You did not attempt to address about the longer hospitalization which caught my attention. Was there a reason why not? I do not encourage you to address too much about those potentially beneficial but statistically insignificant effect. The explanation that longer hospitalization in probiotic group might be due to the lower discharge weight does not make too much of sense to me. As there was no difference in the initial birth weight and GA, the your explanation might indicate that probiotics slows down the weight gain.
(Reviewer #3, comment #4)
We agree with the reviewer that generally none statistically significant effects should not be addressed to a large extent. However, as feeding tolerance and growth where our primary outcomes, presenting these in in the summary of the main results was important.
We believe that we have addressed all the primary and secondary outcomes, both with positive and negative response to probiotics. In line 541 to 550 in the summary of the main results, we present the non-beneficial results including poorer weight and length gain as well as hospitalization.
In our opinion, lower weight at discharge is one credible explanation to why the hospitalization is longer in the probiotics group. These infants are feed with a feeding tube until breastfeeding or bottle feeding can be fully established. Whereas some clinics can discharge patients still in need of feeding tube with continued homecare and follow-up visits, many patients are hospitalized until enteral feeds can be discontinued. One important indicator for when enteral feeds can be stopped is the infants weight.
Another potential explanation could be that other neonatal morbidity such as bronchopulmonary dysplasia are more common in the probiotics group, ultimately increasing the need for longer hospital care. However, as we have no data on this, we do not wish to speculate.
- Some abbreviations, such as Nct, Ctri, and ChiCtr, need clarification. They appear both in the main text and supplementary material without providing any information.
(Reviewer #3, comment #5)
Thank you for pointing this out. Unfortunately, in the manuscript these abbreviations are part of a link to trial registrations in Table 2 and cannot be removed or altered for then the link will be invalid.
In supplemental file 1, we have renamed these to the full name of the trial registry in column A.

Round 2
Reviewer 3 Report
Comments and Suggestions for Authors
I have no more comments.